# The Requirement of Glycoprotein C for Interindividual Spread Is Functionally Conserved within the Alphaherpesvirus Genus (*Mardivirus*), but Not the Host (*Gallid*)

**DOI:** 10.3390/v13081419

**Published:** 2021-07-21

**Authors:** Widaliz Vega-Rodriguez, Nagendraprabhu Ponnuraj, Maricarmen Garcia, Keith W. Jarosinski

**Affiliations:** 1Department of Pathobiology, College of Veterinary Medicine, University of Illinois at Urbana-Champaign, Urbana, IL 61802, USA; widaliz2@illinois.edu (W.V.-R.); naguponu@illinois.edu (N.P.); 2Poultry Diagnostic and Research Center, Department of Population Health, College of Veterinary Medicine, University of Georgia, Athens, GA 30602, USA; mcgarcia@uga.edu

**Keywords:** herpesvirus, glycoprotein C, interindividual spread, Marek’s disease, *Gallid*, *Meleagrid*

## Abstract

Marek’s disease (MD) in chickens is caused by *Gallid alphaherpesvirus* 2, better known as MD herpesvirus (MDV). Current vaccines do not block interindividual spread from chicken-to-chicken, therefore, understanding MDV interindividual spread provides important information for the development of potential therapies to protect against MD, while also providing a natural host to study herpesvirus dissemination. It has long been thought that glycoprotein C (gC) of alphaherpesviruses evolved with their host based on their ability to bind and inhibit complement in a species-selective manner. Here, we tested the functional importance of gC during interindividual spread and host specificity using the natural model system of MDV in chickens through classical compensation experiments. By exchanging MDV gC with another chicken alphaherpesvirus (*Gallid alphaherpesvirus* 1 or infectious laryngotracheitis virus; ILTV) gC, we determined that ILTV gC could not compensate for MDV gC during interindividual spread. In contrast, exchanging turkey herpesvirus (*Meleagrid alphaherpesvirus* 1 or HVT) gC could compensate for chicken MDV gC. Both ILTV and MDV are *Gallid alphaherpesviruses*; however, ILTV is a member of the *Iltovirus* genus, while MDV is classified as a *Mardivirus* along with HVT. These results suggest that gC is functionally conserved based on the virus genera (*Mardivirus* vs. *Iltovirus*) and not the host (*Gallid* vs. *Meleagrid*).

## 1. Introduction

All avian herpesviruses are members of the *Alphaherpesvirinae* within the *Herpesviridae* [1] and include *Gallid alphaherpesviruses* (GaHV) 1, 2, and 3 and *Meleagrid alphaherpesvirus* 1 (MeHV-1), better known as infectious laryngotracheitis virus (ILTV) or GaHV-1, Marek’s disease herpesvirus (MDV) or GaHV-2, GaHV-3, and turkey herpesvirus (HVT), respectively. MDV, GaHV-3, and HVT are classified into the genus *Mardivirus*, while ILTV is classified as an *Iltovirus* based on genomic sequencing [2].

ILTV (GaHV-1) is the prototypic member of the *Iltovirus* genus that also includes *Psittacid alphaherpesvirus* 1 (PsHV-1) [2]. ILTV is highly contagious and results in significant losses to the poultry industry due to severe respiratory disease including conjunctivitis, sinusitis, oculo-nasal discharge, bloody mucus, and overall high morbidity [3]. For the most part, ILTV is localized to the respiratory tract and interindividual spread is through shedding of respiratory secretions and transmitted by inhalation of infectious material. Similarly, MDV and HVT infection is initiated in the lungs of chickens or turkeys, respectively. However, infection in the respiratory system is initiated by inhalation of infectious material in dander and dust that contains infectious virus previously shed from feather follicle (FF) epithelial (FFE) skin cells of infected birds. The most well studied *Mardivirus* is MDV where transmission of MDV can be through direct bird-to-bird contact or through indirect contact with infected feathers, dust, or dander. It is not completely understood what mechanism is used by MDV to spread in a flock; however, MDV glycoprotein C (gC) is essential for interindividual spread [4,5,6,7]. The requirement of gC for interindividual spread was recently confirmed to be conserved for another *Mardivirus*, GaHV-3 [8] showing the functional importance of gC in *Mardivirus* interindividual spread. To date, only shedding of infectious virus using a gC-null ILTV was shown to be attenuated in vivo, but interindividual spread was never directly addressed [9].

MDV gC, previously referred to as the “A-antigen”, is encoded by the UL44 gene in the MDV genome, and it is conserved among the *Alphaherpesvirinae*. MDV gC is highly expressed during in vitro and in vivo propagation; however, its expression is reduced following serial passage in tissue culture cells [5,10,11,12,13,14,15]. Alphaherpesvirus gC proteins have been shown to be important for multiple functions during herpesvirus infection including primary attachment of cell-free virus to heparin sulfate- and chondroitin-like glycosaminoglycans on the surface of cells [16,17] and the late steps of egress from cultured cells [16,18]. gC homologs have also been shown to have immune evasion functions. For example, herpes simplex virus 1 (HSV-1), HSV-2, and *Suid alphaherpesvirus* 1 (SuHV-1) or pseudorabies virus (PRV), *Equid alphaherpesvirus* 1 (EHV-1), *Saimiriine alphaherpesvirus* 1 (SaHV-1), and *Bovid alphaherpesvirus* 1 (BoHV-1) gC proteins are thought to have immune evasive functions by binding and inhibiting the complement components C3 [19,20,21,22,23,24,25].

Following numerous studies examining *Mardivirus* gC proteins during interindividual spread in chickens [4,5,6,7,8,26], we asked whether other alphaherpesvirus gC proteins can compensate for MDV gC during interindividual spread. It has long been thought that gC proteins evolved with the host based on the species-selective interaction of different alphaherpesvirus gC proteins and complement C3 in which the ability to bind and inhibit C3 was conserved within the virus and host genera [24]. That is, HSV-1 and -2 gC efficiently bound and inhibited human C3 but not equine or chicken C3. To test the hypothesis that gC proteins evolved with the host, we tested the ability of ILTV (GaHV-1) and HVT (MeHV-1) gC to compensate for GaHV-2 (MDV) gC during interindividual spread. To perform this, we replaced MDV gC with ILTV or HVT gC and tested the ability of each virus to spread from chicken to chicken. Interestingly, MDV expressing ILTV gC was unable to spread, while MDV expressing HVT gC efficiently spread from chicken to chicken. Collectively, these results show that the function of gC during interindividual spread is conserved among the virus genera (*Mardivirus* vs. *Iltovirus*), but not within the host (*Gallid* vs. *Meleagrid*). Our results suggest the evolution of avian gC was, at least partially, based on the pathogenesis of the virus and not through evolution with the host.

## 2. Materials and Methods

### 2.1. Cell Culture and Cells

Chick kidney cells (CKCs) were prepared from 2–4 weeks-old specific-pathogen-free (SPF) chickens, obtained from the University of Illinois at Urbana-Champaign (UIUC) Poultry Farm, following standard methods [27] and seeded in growth medium consisting of Medium 199 (Cellgro, Corning, NY, USA) supplemented with 10% tryptose-phosphate broth (TPB), 0.63% NaHCO_3_ solution, and antibiotics (100 U/mL penicillin and 100 µg/mL streptomycin) and 4% fetal bovine serum (FBS). Confluent CKCs were maintained in F10.199 medium consisting of a 1:1 mixture of Ham’s F10 (Cellgro) and Medium 199 supplemented with 7.5% TPB, 0.63% NaHCO_3_, 0.2% FBS, and antibiotics.

Chicken embryo cells (CECs) were prepared from 10–11-day-old SPF embryos obtained from the UIUC Poultry Farm following standard methods [27]. Briefly, primary CECs were seeded in growth medium consisting of Medium 199 supplemented with 10% TPB, 0.63% NaHCO_3_ solution, antibiotics, and 4% FBS. Confluent CECs were maintained in Medium 199 supplemented with 7.5% TPB, 0.63% NaHCO_3_, 0.2% FBS, and antibiotics.

The chicken DF-1-Cre fibroblast cell line [28] was cultivated in a 1:1 mixture of Leibovitz L-15 and McCoy 5A (LM) media (Gibco, Gaithersburg, MD, USA) supplemented with 10% FBS and antibiotics, and maintained in 50 µg/mL Zeocin (Invitrogen, Carlsbad, CA, USA).

All cells were maintained at 38 °C in a humidified atmosphere of 5% CO_2_.

### 2.2. Generation of Recombinant (r)MDVs

To clone HVT gC, the HVT UL44 ORF was cloned from a previously described HVT bacterial artificial chromosome (BAC) clone [29] into pcDNA3.1 (Invitrogen) using standard techniques. Briefly, primers overlapping the start and stop codon were designed with HindIII and XbaI sites on the ends and amplified by PCR using Dream Taq PCR Master Mix (Thermo Fisher Scientific, Waltham, MA, USA) using primers shown in Table 1, gel purified, and cloned into the multiple clone site of pcDNA3.1. The ILTV expression construct has been previously described [30].

To generate HVT and ILTV gC transfer vectors, the I-SceI-AphAI cassette from pEP-KanSII was amplified by PCR with Phusion Flash High-Fidelity PCR Master Mix (Thermo Fisher Scientific) using primers shown in Table 1 and inserted into the coding sequence of HVT and ILVT gC using BamHI or AflII, generating pcHVTgC-in and pcILTVgC-in, respectively. To generate pcHVTgC with a C-terminal Myc-His tag, Gibson assembly reaction mixture (New England Biolabs, Inc., Ipswich, MA, USA) was used to insert the Myc-His sequence into pcHVTgC-in to generate pcHVTgC-MycHis-in (pcHVTgC*-in) using primers in Table 1. Subsequently, the gC proteins were amplified from their respective transfer vectors by PCR with Phusion Flash High-Fidelity PCR Master Mix using primers shown in Table 2 and inserted into rΔgC [7] for mutagenesis in GS1783 *Escherichia coli* cells. Restriction fragment length polymorphism (RFLP) analyses, analytical PCR, and DNA sequencing confirmed all clones were correct. Primers used for sequencing have been previously published [6,7].

rMDVs were reconstituted by transfecting DF-1-Cre cells, which efficiently remove the mini-F BAC sequences from the viral genome [28], with purified BAC DNA plus Lipofectamine 2000 (Invitrogen) using the manufacturers’ instructions. Transfected DF-1-Cre cells were mixed with fresh primary CKCs or CECs until plaques formed, then further propagated in CKCs or CECs until virus stocks could be stored. All rMDVs were used at ≤5 passages for in vitro and in vivo studies.

### 2.3. Measurement of Plaque Areas

Plaque areas were measured as previously described [31]. Briefly, CECs were seeded in 6-well dishes and infected with 100 plaque-forming units (PFU) per well. After 5 days, cells were washed once with phosphate buffered saline (PBS), fixed and permeabilized with PFA buffer (2% paraformaldehyde, 0.1% Triton X-100) for 15 min, and washed twice with PBS. Immunofluorescence assays (IFAs) were performed as previously described [31] using anti-MDV chicken sera and goat anti-chicken IgY-Alexa Fluor^®^ 568 or 488 secondary antibody (Molecular Probes, Eugene, OR). Digital images of 50 individual plaques were collected using an EVOS FL Cell Imaging System (Thermo Fisher Scientific) and compiled using Adobe Photoshop 21.0.1 release. Plaque areas were measured using ImageJ [32] version 1.51k software, and means were determined for each plaque population. box and whisker plots were generated using Microsoft^®^ Excel^®^ for Microsoft 365.

### 2.4. Viral Replication Kinetics in Cell Culture

To measure viral replication kinetics of viruses in cell culture, qPCR assays were used to measure the relative level of replication as previously described [31]. Briefly, CECs were prepared in 6-well tissue culture plates and the next day inoculated with 100 PFU/well. Total DNA was collected from the inoculum and at 0, 24, 48, 72, 96, and 120 h following infection, using the QIAamp DNA Mini Kit (Qiagen, Germantown, MD, USA). Quantification of MDV genomic copies in CECs was performed using primers and probe to MDV ICP4 and chicken iNOS in duplex reactions as previously described [31,33]. All qPCR assays were performed as absolute quantification using standard curves in an Applied Biosystems QuantStudio 3 Real-Time PCR System (Thermo Fisher Scientific, Waltham, MA, USA) and the results were analyzed using the QuantStudio Design & Analysis Software v1.4.2. The coefficient of regression was >0.99 for standard curves.

### 2.5. Animal Experiments

In Trial 1, commercial Pure Columbian (PC) × New Hampshire (NH) cross chickens were used, while in Trial 2, PC chickens were used. All birds were obtained from the UIUC Poultry Farm (Urbana, IL) and were from MD-vaccinated parents and considered maternal antibody positive at hatch.

In Trial 1, five-day old PC × NH chicks were experimentally infected by intra-abdominal inoculation of 2000 PFU for each rMDV and housed in separate rooms (*n* = 6/group). For each group, another group of chickens (*n* = 11/group) were left uninfected to act as contact controls to determine whether rMDVs were able to interindividual spread. In Trial 2, seven-day old PC chickens were infected as in Trial 1 (*n* = 10/group) and housed with uninfected contact chickens (*n* = 9/group). In this trial, experimentally infected birds were sacrificed to collect tissue samples, therefore, only 4–7 chickens were available to measure MD incidence. For both trials, chickens were evaluated daily, euthanized when birds showed clinical signs of MD (e.g., lethargy, depression, paralysis, and torticollis), and examined for gross MD lesions. Chickens positive for MD included birds succumbing to disease prior to the experimental termination date and birds positive for MD-related lesions at termination of the experiment. 

### 2.6. Viral Replication Kinetics In Vivo

Whole blood was collected by wing-vein puncture [4] at different time points and DNA was extracted using the E.Z. 96 blood DNA kit from Omega Bio-tek, Inc. (Norcross, GA) using the manufacturer’s instructions. Quantification of MDV genomes in blood was performed exactly as described for viral replication kinetics in cell culture.

### 2.7. Monitoring rMDVs in Feather Follicles (FFs)

To monitor the time at which each rMDV reached the FFs, two flight feathers were plucked from the right and left wings (4 total) of experimentally-infected birds weekly starting at 7 days post-infection (pi) for pUL47eGFP expression. A Leica M205 FCA fluorescent stereomicroscope with a Leica DFC7000T digital color microscope camera (Leica Microsystems, Inc., Buffalo Grove, IL, USA) was used to document pUL47eGFP expression. Feather plucking for all experimentally infected birds in Trial 2 was discontinued after 42-days pi because only a few experimentally infected birds remained. 

### 2.8. Immunofluorescence Assay (IFA) of FFs

Whole feathers plucked from chickens infected with different rMDVs were fixed using PFA buffer, washed twice with PBS, and then blocked in 10% neonatal calf serum. Fixed FFs were stained with primary anti-gC monoclonal A6 antibody [7], -ILTV gC monoclonal mAb8 antibody [30], or -Myc (Sigma-Aldrich, St Louis, MO, USA), and anti-mouse Ig Alexa Fluor 568 (Molecular Probes, Eugene, OR, USA) was used as secondary antibody. Digital images were taken with Leica DFC7000T digital color microscope camera mounted on Leica M205 FCA fluorescent stereomicroscope. All images were compiled using Adobe Photoshop 21.0.1 release.

### 2.9. Statistical Analyses 

IBM SPSS Statistics Version 27 software (SPSS Inc., Chicago, IL, USA) was used for statistical analyses. Plaque size assays were analyzed using one-way analysis of variance (ANOVA) with virus included as fixed effect and the plaque size used as a dependent variable. The normalized data for viral replication (qPCR) were analyzed using two-way ANOVA followed by LSD and Tukey’s post hoc tests; virus (V) and time (T) and all possible interactions (V × T) were used as fixed effects. The genomic copies were the dependent variable. Fisher’s exact tests were used for infection and transmission experiments. Statistical significance was declared at *p* < 0.05. Mean tests experiments associated with significant interaction (*p* < 0.05) were separated using Tukey’s post hoc test.

## 3. Results

### 3.1. Generation and Characterization of rMDVs

#### 3.1.1. Generation of rMDV Expressing ILTV or HVT gC

To generate rMDV expressing either ILTV or HVT gC, the coding sequence of both ILTV and HVT UL44 open reading frames (ORFs) were inserted into rΔgC, previously described [7], using two step Red-mediated recombination. Figure 1a shows a schematic representation of rWT, rΔgC, and the newly generated rILTVgC, rHVTgC, and rHVTgC*. Restriction fragment length polymorphism (RFLP) analysis of rWT, rΔgC, rILTVgC, rHVTgC, and rHVTgC* confirmed the integrity of the BAC clones as the predicted banding patterns were observed (Figure 1b). PCR and DNA sequencing were used to confirm that each clone was correct at the nucleotide level (data not shown) using previously published primers flanking MDV UL44 [4,6].

#### 3.1.2. Replication of rMDV Expressing ILTV or HVT gC in Cell Culture

Following reconstitution of each virus in CECs, plaque size assays were performed to measure the ability of the rMDVs to replicate in cell culture (Figure 2a). As it has been previously described, vΔgC generated significantly larger plaques than vWT [5,7,14]. Interestingly, vILTVgC and vHVTgC also generated significantly larger plaques than vWT. Addition of the C-terminal tag on HVTgC (vHVTgC*) resulted in reduction of plaque sizes similar to vWT. Multi-step viral replication kinetics showed that vWT was significantly different to all other viruses at 24 h pi. At 120 h pi, vWT and vHVTgC were both significantly different to vILTVgC and vHVTgC*, and vHVTgC was significantly different to vΔgC. vΔgC was significantly different to all viruses at this timepoint (Figure 2b). In all, these results are consistent with the inhibitory effect MDV gC has on MDV cell-to-cell spread and suggests ILTV and HVT gC do not have this inhibitory effect on MDV replication.

### 3.2. Characterization of rMDVs in Experimentally Infected Chickens

#### 3.2.1. Replication of rMDVs during Experimental Infection

Next, we tested the ability of vILTVgC, vHVTgC, and vHVTgC* to replicate in experimentally infected chickens. To measure in vivo replication, we determined the MDV genomic copies in blood from experimentally infected chickens over 35 (Trial 1) and 21 (Trial 2) days pi. Results showed that vΔgC was significantly different to all viruses at all time points except for 3- and 35-days pi in Trial 1, while vWT was significantly different to all viruses at 35 days pi (Figure 3a). In Trial 2, no significant differences were measured between all viruses at the same time point (Figure 3b). Since each rMDV expressed pUL47eGFP [13,14,15,16,17,18,19], we monitored the ability of each virus to infect FFE cells required for interindividual spread. To perform this, feathers were plucked from each experimentally infected bird and expression of pUL47eGFP was visualized using a fluorescent microstereoscope. There were no significant differences in the total number of birds that were infected in each group ranging from 100% for vΔgC to 83% positive for all other groups in Trial 1 (Figure 3c). In Trial 2, pUL47eGFP positivity ranged from 89% for vHVTgC to 50% for vHVTgC* (Figure 3d).

#### 3.2.2. MD Induction of rMDVs during Experimental Infection

There were no significant differences in MD incidence among the experimentally infected groups in both trials (Figure 3e,f). At termination of the experiment, blood cells and serum were collected from all birds that remained negative for pUL47eGFP expression in FFs and did not develop MD. Blood cells were used in qPCR assays to detected viral genomes and serum was used to detect anti-MDV antibodies and all were negative for both assays. Figure 3g,h summarize all data combined showing all but one experimentally infected chicken in the vILTVgC group in Trial 2 were infected. The results show that MDV, ILTV, and HVT gC play no role in virus replication and MD induction in experimentally infected chickens.

#### 3.2.3. Expression of gC proteins in FFs

To determine whether rMDV expressed their respective gC proteins following experimental infection of chickens, feathers plucked during the course of the experiment were stained with specific antibodies. FFs obtained from Trial 1 showed positive staining for MDV gC in the vWT-infected FFs using anti-MDV gC antibody, while vΔgC, vILTVgC, and vHVTgC were negative for MDV gC (Figure 4a). Using anti-ILTV gC antibody, positive staining was observed from vILTVgC-infected FFs showing vILTVgC maintained its expression during in vivo replication (Figure 4b). Since we did not have an antibody specific for HVT gC, we utilized the C-terminal Myc-His tag on HVT gC to confirm HVTgC* expression was maintained in vHVTgC*-infected FFs in Trial 2 (Figure 4c). These results confirm each rMDV maintained expression of their respective gC protein during experimental infection in chickens. 

### 3.3. gC Proteins Are Functionally Conserved among the Mardivirus and Not the Gallid Genera

#### Natural Infection with rMDVs Expressing ILTV or HVT gC

To determine whether each rMDV could interindividual spread in chickens, both infection in FFs and MD incidence in naturally infected (contact) chickens housed with the experimentally infected groups was determined. No contact chickens were infected in the vΔgC-Con group based on the presence of MDV in the FFs (Figure 3c) and MD incidence (Figure 3e) that is consistent with the requirement of MDV gC for interindividual spread [4,5,7]. Interestingly, no contact chickens in the vILTVgC-Con group were naturally infected based on pUL47eGFP in the FFs (Figure 3c,d) and MD incidence (Figure 3e,f), while contacts in the vHVTgC-Con (Trials 1 and 2) and vHVTgC*-Con (Trial 2) groups were infected. To confirm contacts negative for pUL47eGFP in FFs or MD were never infected with MDV, qPCR assays for viral DNA in the blood and IFAs for anti-MDV antibodies in their serum were used and these birds were confirmed to be negative for infection (data not shown). A summary of all naturally infected birds is shown in Figure 3g,h. All contact birds in the vΔgC-Con (Trial 1) and vILTVgC-Con (Trials 1 and 2) groups were negative for infection. These data show that both HVT gC and HVT gC with a C-terminal Myc-His tag (vHVTgC*) were able to facilitate natural infection of MDV, while ILTV gC (vILTVgC) was unable to compensate for MDV gC. We can conclude that turkey HVT gC is functionally conserved with chicken MDV gC, while chicken ILTV gC is not conserved.

## 4. Discussion

It has been shown that alphaherpesvirus gC proteins bind and inhibit complement C3 through species-selective interactions based on their ability to bind to C3 from their respective hosts [24]. Thus, it has been suggested the gC proteins evolved with their respective hosts. Utilizing our model for examining the essential role gC proteins play during interindividual spread in chickens [3,4,5,6,7,25] we sought to test the specificity of avian gC proteins. To perform this, we exchanged chicken MDV gC with chicken ILTV gC or turkey HVT gC and tested the ability of each gC protein to compensate for MDV gC in MDV interindividual spread. Our results conclusively showed that MDV expressing ILTV gC was defective during interindividual spread in chickens, while MDV expressing HVT gC readily spread. Both MDV and ILTV are *Gallid alphaherpesviruses*, but belong to the *Mardivirus* and *Iltovirus* genera, respectively. HVT is a *Meleagrid alphaherpesvirus* but is characterized in the *Mardivirus* genus with MDV. These results show gC proteins did not necessarily evolve with the host in a species-selective mechanism-based chicken ILTV gC being unable to compensate for chicken MDV gC, while turkey HVT gC was able to compensate for MDV gC during interindividual spread.

Former studies on MDV plaque sizes generated from gC-null viruses showed significantly increased plaque sizes [4,6,12]. The mechanism for the increased replication is not understood. Interestingly, MDV expressing ILTV or HVT gC also generated increased plaque sizes (Figure 2a) suggesting both proteins lack the inhibitory function that MDV gC possesses in cell culture. Multi-step replication kinetics also showed differences during cell culture propagation, particularly increased replication of vΔgC at 120 h pi, the same time at which plaque sizes were measured (Figure 2b). Interestingly, no differences were seen in the vILTVgC and vHVTgC groups that conflicts with the plaque size assay data. The reason for this discrepancy is not known but could be due to the sensitivity of each assay and the output being measured. Since one-step replication kinetics are not possible with cell-associated MDV, we are limited in the assays to measure virus replication in cell culture to identify potential replication defects. However, in all, there were no replication defects for all viruses in cell culture and only increased replication compared to vWT.

Currently, it has not been directly shown whether avian herpesviruses bind and inhibit complement C3; however, the *Mardiviruses* MDV, GaHV-3, and HVT gC proteins encode the conserved cysteines predicted to be important for folding and binding C3 [34]. In contrast, ILTV gC does not encode all these cysteines, where only C1-C2 and C5-C8 are present. Figure 5 shows alignment and predicted motifs identified between the MDV, GaHV-3, HVT, and ILTV gC proteins including the conserved cysteines. It can be predicted these differences result in altered structural features between MDV and ILTV gC folding and helps to explain the lack of compensation for MDV gC during interindividual spread of MDV. We have formerly shown MDV gC can compensate for GaHV-3 gC [8] in chickens, and in the present report, we show that HVT gC can compensate for MDV gC during interindividual spread (Figure 3). Although all combinations of gC compensation experiments between the viruses is not complete, the current data strongly suggests highly conserved functions for gC proteins between these three *Mardiviruses*. Further experiments determining whether avian gC proteins bind and inhibit C3 is warranted to address these differences and the potential role gC and complement may play during natural infection in the host.

Despite encoding the Marek_A Superfamily and Ig-like domains that most gC proteins in the conserved domain database (CDD) [36] have, ILTV gC is only 15% conserved to MDV gC (Table 3). In retrospect, based on the protein sequence homology between MDV and ILTV gC (Figure 5), it is not a surprise that ILTV gC would not compensate for MDV gC and undoubtedly has significantly different functions compared to MDV gC. Interestingly, turkey HVT gC is more conserved to MDV gC (75.7%) than chicken GaHV-3 that is 72.7% identical MDV. Although there is only a small difference between HVT and GaHV-3 gC protein identities to MDV gC (75.7 to 72.7%), this is further evidence that the alphaherpesvirus gC proteins did not necessarily evolve with the host. It is likely the gC protein of ILTV evolved very early during the divergence of the *Gallid alphaherpesviruses* and possibly MDV, GaHV-3, and HVT evolved after this divergence. These results also suggest the potential cellular targets for MDV, GaHV-3, and HVT gC proteins may be conserved between *Gallid* and *Meleagrid* host genera.

In conclusion, this report suggests the functional conservation of gC during interindividual spread is not conserved within the host but is conserved within the virus genus and host specificity of MDV in chickens. Future studies in our laboratory are directed at understanding the conserved and divergent functions of gC proteins in the context of natural infection to better understand the evolution of host specificity and disease pathogeneses.

## Figures and Tables

**Figure 1 viruses-13-01419-f001:**
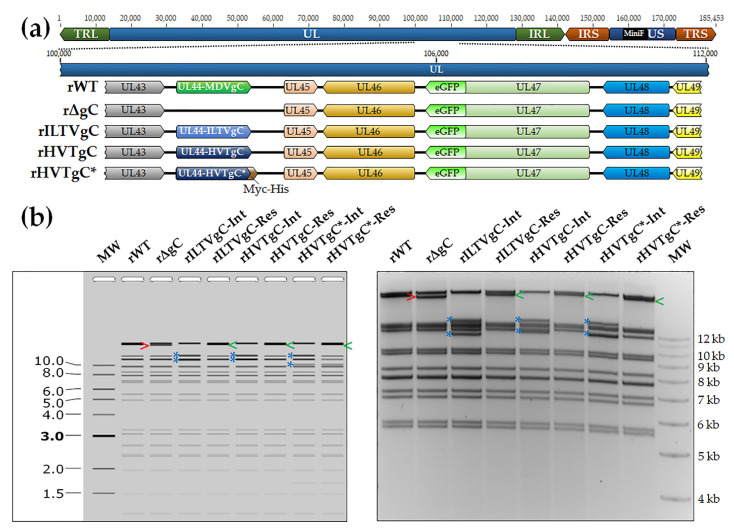
Generation of rMDVs. (**a**) The MDV genome in schematic form (not to scale) depicting the locations of the terminal (TRL) and internal repeat long (IRL), terminal (TRS) and internal repeat short (IRS), and unique long (UL) and unique short (US) regions. A portion of the UL region is expanded to show the relevant genes within this region including UL44 and UL47 (eGFP). Differences between the rMDVs are shown with changes in color representing ILTV UL44 (gC) and HVT UL44 in the MDV UL44 locus. rΔgC shows deletion of MDV UL44 (gC). (**b**) Predicted RFLP diagram was generated using SnapGene software (from Insightful Science; available at snapgene.com) and is shown in the left panel. BAC DNA obtained for rWT (MDV gC), rΔgC (ΔMDV gC), rILTVgC, rHVTgC, and rHVTgC* integrate and resolved clones were digested with HindIII and used in RFLP analysis. For the generation of rILTVgC, insertion of the AphAI cassette into the HindIII fragment incorporates two additional HindIII sites resulting in the reduction of the HindIII fragment (>) from 23,369 bp to 13,915 (*), and 11,660 bp (*). Following resolution, removal of the AphaI cassette results in a 24,537 bp HindIII site (<). For the generation of rHVTgC, insertion of the AphAI cassette into the HindIII fragment incorporates two additional HindIII sites resulting in 13,997 (*) and 12,003 bp (*) fragments. Following resolution, removal of the AphaI cassette results in a 24,967 bp HindIII site (<). For the generation of rHVTgC*, insertion of the AphAI cassette into the HindIII fragment reduces the HindIII fragment (>) from 23,369 bp to 13,997 (*) and 11,937 bp (*) fragments. Following resolution, removal of the AphAI cassette results in a 24,885 bp HindIII site (<). The molecular weight marker used was the GeneRuler 1 kb Plus DNA Ladder from Thermo Scientific (Carlsbad, CA, USA).

**Figure 2 viruses-13-01419-f002:**
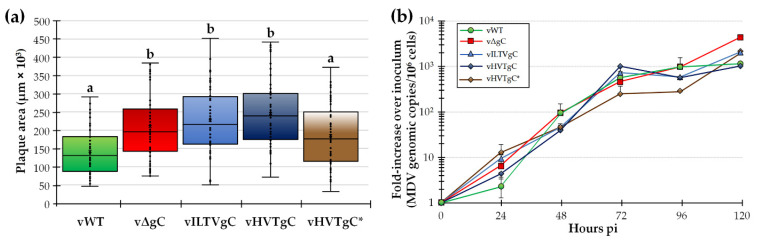
Replication of rMDVs in tissue culture cells. (**a**) Mean plaque areas for viruses reconstituted from BAC clones described in Figure 1 were measured and shown as box and whisker plots. Significant differences were determined using one-way ANOVA (*p* < 0.05, *n* = 250). Mean plaque areas with different letters are significantly different using LSD and Tukey’s post hoc tests (*p* ≤ 0.05). (**b**) Multi-step replication kinetics was used to measure virus replication in CECs. Total viral genome copies were measured for each virus at 24, 48, 72, 96, and 120 h pi. Shown is the fold-increase over inoculum at day 0. vWT was significantly different to all other viruses at 24 h pi. At 72 h pi, vHVTgC was significantly different to vΔgC and vHVTgC*, while vHVTgC* was significantly different to vHVTgC. At 96 h pi, vWT and vΔgC were significantly different to vHVTgC*, and vHVTgC* was significantly different to both, vWT and vΔgC, as well. At 120 h pi, vWT and vHVTgC were both significantly different to vILTVgC and vHVTgC*. vΔgC was significantly different to all viruses at this timepoint (*p* < 0.05, two-way ANOVA, LSD and Bonferroni, *n* = 3).

**Figure 3 viruses-13-01419-f003:**
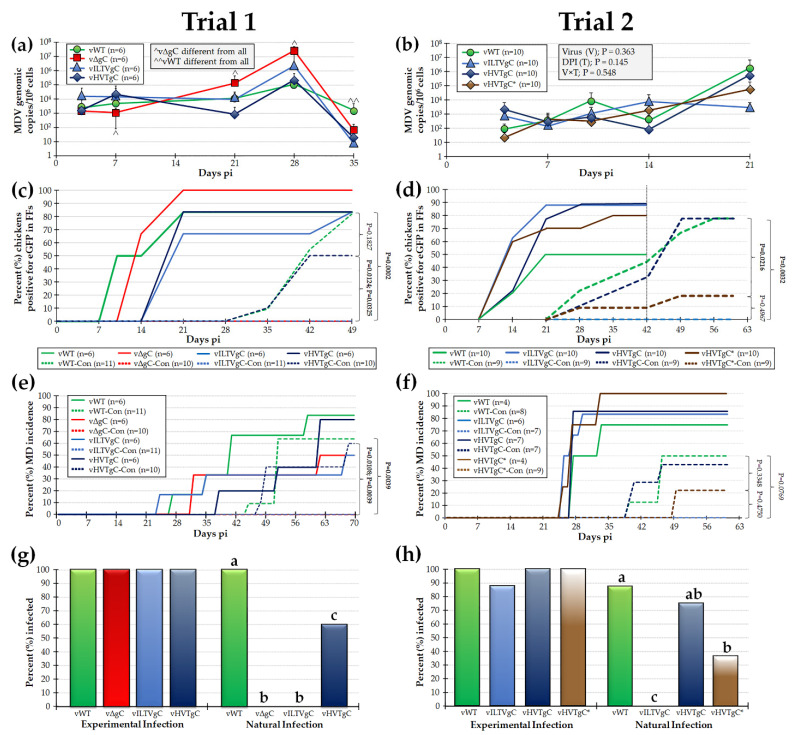
Replication of rMDVs in experimentally and naturally infected chickens. PC × NH (Trial 1) and PC (Trial 2) chickens were inoculated with vWT, vΔgC, vILTVgC, vHVTgC, or vHVTgC* as described in the Materials and Methods for 70 (Trial 1) or 63 (Trial 2) days. (**a**,**b**) MDV replication was monitored in experimentally infected chickens by quantification of MDV genomes in the blood over the first 3–5 weeks of infection. Shown are the mean MDV genomic copies per 10^6^ blood cells ± standard deviations. Only vΔgC was significantly different to all other viruses at 7, 21, and 28 days pi, while vWT was significant to all other viruses at 35 days pi in Trial 1 (*p* < 0.05). (**b**) In Trial 2, no significant differences (*p* > 0.05, *n* = 109) were determined between all viruses at the same time points. (**c**,**d**) Quantitative analysis of the percent of birds positive for pUL47eGFP in FFs over the course of the experiment. Using Fisher’s exact test at *p* < 0.05, there was no significant difference in the total number of chickens positive for experimentally infected chickens in both Trials. No naïve contact chickens housed with vΔgC (Trial 1) or vILTVgC (Trials 1 and 2) were naturally infected, while all other viruses were able to infect contact chickens. *p* values for naturally infected chickens are shown using Fisher’s exact tests and bolded if significant. (**e**,**f**) Total MD incidence was determined by identification of gross lesions in dead or euthanized chickens. There were no significant differences in the total number of chickens developing MD in experimentally infected chickens in both Trials. *p* values for naturally infected chickens are shown using Fisher’s exact tests. (**g**,**h**) The total number of chickens infected based on viral genomes in the blood, pUL47eGFP positivity in FFs, MD, and anti-MDV antibodies in serum when all data was combined. Fisher’s exact tests determined there were no significant differences between all rMDVs in experimentally infected birds. Both vΔgC (Trial 1) and vILTVgC (Trials 1 and 2) were unable to spread to contact chickens. Groups with different letters are significantly different using Fisher’s exact tests (*p* ≤ 0.05).

**Figure 4 viruses-13-01419-f004:**
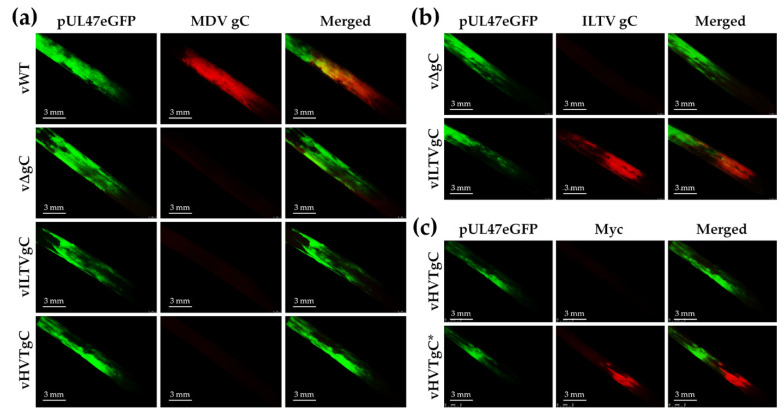
gC protein expression in experimentally infected chickens. Feathers were plucked from vWT, vΔgC, vILTVgC, vHVTgC, and vHVTgC* at 28 days pi from both experimental infection trials. Representative FFs were fixed, then stained using anti-MDV gC (**a**), -ILTV gC (**b**), or -Myc (**c**) antibodies. Expression of pUL47eGFP was used to identify infected FFs. All feathers are orientated with the outer vane of the feather (not shown) towards the left of the image.

**Figure 5 viruses-13-01419-f005:**
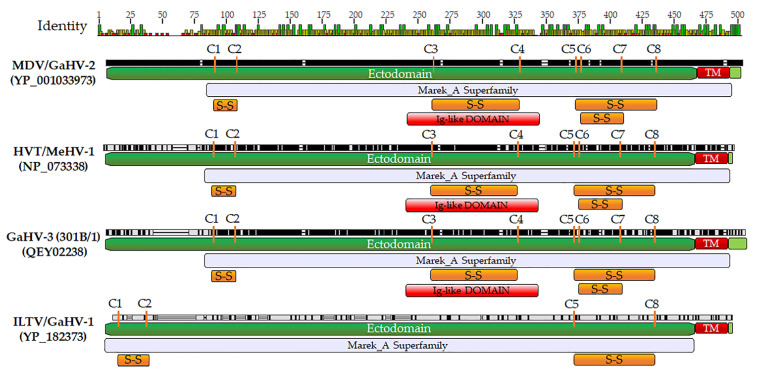
Alignment of MDV (YP_001033973; RB-1B), HVT (NP_073338; FC126), GaHV-3 (QEY02238; 301B/1), and ILTV (YP_182373; USDA ref) gC proteins using MUSCLE Alignment in Geneious Prime 2021.0.3 (Biomatters, Inc., San Diego, CA). The predicted signal sequence was removed from each protein before alignment that represents the ectodomain, transmembrane domain (TM), and short cytoplasmic domain. The eight cysteines predicted to be important for disulfide binding (S-S) and folding are shown, as well as predicted motifs for each protein using MyHits motif scan [35].

**Table 1 viruses-13-01419-t001:** Primers used for cloning genes and generation of shuttle vectors.

Construct ^1^	Direction ^2^	Sequence (5′→3′) ^3^
pcHVTgC	For	CGTAAGCTTTGTGTTTTATTGAGCGGTCG
	Rev	CGTTCTAGATTTGGCCGCTGCGTGATACC
pcHVTgC-in	For	CGACGGGATCCCCAGGGTTCTTTCTGGACTAGTCCTACACCCCGTGGAAATAGGGATAACAGGGTAATCGATTT
	Rev	TTAACGGATCCGCCAGTGTTACAACCAATTAACC
pcILTVgC-in	For	TCGCACTTAAGTGTTGAAGCGCTTGGCGCTTATCCTCCATCTGCTGCGCTGGGTATAGGGATAACAGGGTAATCGATTT
	Rev	TTAACCTTAAGGCCAGTGTTACAACCAATTAACC
pcHVTgC*-in	Vector For	TGAGTTTAAACCCGCTGATCGTTTAAACCCGCTGATCAGCCT
	Vector Rev	TCGAAGGGCCCTCTAGACTCATTCCGCCCCGGTAGG
	Insert For	TTTACCTACCGGGGCGGAATGAGTCTAGAGGGCCCTTCGAACAAAA
	Insert Rev	GCTGATCAGCGGGTTTAAACGATCAGCGGGTTTAAACTCAATGGT

^1^ Construct generated with the set of primers. ^2^ Directionality of the primer. ^3^ Underlined sequences indicated restriction enzymes used in the cloning. Bold nucleotides indicate priming sites within the mutagenesis template plasmid pEP-KanSII. *Addition of the Myc C-terminal tag on HVTgC (vHVTgC*).

**Table 2 viruses-13-01419-t002:** Primers used for generation of recombinant Marek’s disease herpesvirus (MDV).

Modification ^1^	Direction ^2^	Sequence (5′→3′) ^3^
MDV-HVTgC	For	ATACTAAACGATGGAGTTGTGTTTTATGAGCGTTGAAAACGATCCACTAGTAACGGCCGCCAG
	Rev	TCACGTTTCTCCACTATTGCATTATTGTCTGACAAATAAAAGCTCTAGCATTTAGGTGACACT
MDV-ILTVgC	For	ATACTAAACGATGGAGTTGTGTTTTATGAGCGTTGAAAACGATCCACTAGTAACGGCCGCCAG
	Rev	TCACGTTTCTCCACTATTGCATTATTGTCTGACAAATAAAAGCTCTAGCATTTAGGTGACACT
MDV-HVTgC*	For	ATACTAAACGATGGAGTTGTGTTTTATGAGCGTTGAAAACTGTGTTTTATTGAGCGGTCG
	Rev	TCACGTTTCTCCACTATTGCATTATTGTCTGACAAATAAACTGATCAGCGGGTTTAAACG

^1^ Modification of the MDV genome using two-step Red recombination. ^2^ Directionality of the primer. ^3^ Bold nucleotides indicate priming sites within the transfer plasmid. * Addition of the Myc C-terminal tag on HVTgC (vHVTgC*).

**Table 3 viruses-13-01419-t003:** Protein identities for Gallid alphaherpesviruses 1, 2, 3, and Meleagrid alphaherpesvirus 1.

Identities ^1^	GaHV-1 (ILTV)	GaHV-2 (MDV)	GaHV-3	MeHV-1 (HVT)
**GaHV-1 (ILTV)**	-	15.2%	18.3%	16.4%
**GaHV-2 (MDV)**	15.2%	-	72.7%	75.7%
**GaHV-3**	18.3%	72.7%	-	69.6%
**MeHV-1 (HVT)**	16.4%	75.7%	69.6%	-

^1^ Percent (%) protein identities between *Gallid alphaherpesvirus* (GaHV) 1 (GaHV-1) or infectious laryngotracheitis virus (ILTV), GaHV-2 or Marek’s disease virus (MDV), GaHV-3, and *Meleagrid alphaherpesvirus* 1 or turkey herpesvirus (HVT) using MUSCLE Alignment in Geneious Prime 2021.20.3.

## Data Availability

Not applicable.

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
