# Peer review of "The Requirement of Glycoprotein C for Interindividual Spread Is Functionally Conserved within the Alphaherpesvirus Genus (Mardivirus), but Not the Host (Gallid)"

_viruses, 2021, doi:10.3390/v13081419_

Round 1

Reviewer 1 Report

This well written manuscript provides valuable information about the functional aspects of avian herpesvirus gC proteins, which could further be utilized for the better understanding of the pathogenicity of the related disease conditions. The methodology and the the interpretation of the obtained results is sound. I highly recommend accepting and publishing this work.

Author Response

Thank you for the kind review.

Reviewer 2 Report

General Comment: 

In this manuscript, the authors investigated the functions of the glycoprotein C (gC) encoded by various gallid alphaherpesviruses. They deleted gC in the genome of the highly oncogenic Marek’s disease virus (MDV) and replaced it by its counterparts of herpesvirus of turkey (HVT) and infectious laryngotracheitis virus (ILTV). The resulting recombinant viruses replicated efficiently in vitro and induced plaques that were even slightly larger than those of wt MDV. Infection of chickens revealed that the recombinant viruses harboring these gC counterparts efficiently replicate in vivo. Strikingly, they observed that HVT gC can complement the loss of MDV gC in interindividual spread, which did not occur with ILTV gC. Overall, this is a well written manuscript and contains exciting information for the field. Only minor changes should be addressed prior to publication.

Minor Points:

  • Line: The classical name is Marek’s disease virus (MDV) not Marek’s disease alphaherpesvirus.
  • The sentence “ILTV and MDV are Gallid alphaherpesviruses; however, ILTV is a member of the Iltovirus genus, while MDV is classified as a Mardivirus along with HVT.” does not belong into the abstract. Line 19-25 should be improved for a better readability.
  • The ethics statement for the animal experiments is missing in the Material and Methods section. Or did this change in the guidelines?
  • Line 177: Viruses do not grow. Chance to “replication kinetics”
  • Figure 1A: The authors should rather show what was deleted in rΔgC than shifting the focus area to the left.
  • Figure 1B: Some RFLPs (especially the intermediates) don’t exactly match the prediction. The authors should consider doing NGS to validate that no secondary mutations contribute to these changes.
  • Figure 4: the authors should indicate the orientation of the feathers. This is important as UL47-GFP is expressed in different sections compared to the gC variants.
  • The resolution of Figure 5 could be improved.

Author Response

Please see attachment. Responses are in bold.
